Refining the marine reptile turnover at the Early–Middle Jurassic transition

http://orcid.org/0000-0002-8808-6747 Fischer Valentin 1 v.fischer@uliege.be
http://orcid.org/0000-0001-8510-0227 Weis Robert 2
Thuy Ben 2
1 Evolution & Diversity Dynamics Lab, UR Geology, Université de Liège , Liège , Belgium
2 Department of palaeontology, Natural History Museum Luxembourg , Luxembourg , Grand-Duchy of Luxembourg
Young Mark
Electronic publication date: 2021 Feb 22
Publication date: 2021
Volume: 9
Electronic Location ID: e10647
Received 2020 Oct 1; Accepted 2020 Dec 4
Copyright: © 2021 Fischer et al.
Copyright year: 2021
Copyright holder: Fischer et al.
License: This is an open access article distributed under the terms of the Creative Commons Attribution License, which permits unrestricted use, distribution, reproduction and adaptation in any medium and for any purpose provided that it is properly attributed. For attribution, the original author(s), title, publication source (PeerJ) and either DOI or URL of the article must be cited.
License URL: https://creativecommons.org/licenses/by/4.0/

Keywords: Ichthyosauria, Ophthalmosauridae, Plesiosauria, Cryptoclididae, Toarcian, Aalenian, Bajocian, Faunal turnover, Extinction

Funding: The authors received no funding for this work.

==============================
Even though a handful of long-lived reptilian clades dominated Mesozoic marine ecosystems, several biotic turnovers drastically changed the taxonomic composition of these communities. A seemingly slow paced, within-geological period turnover took place across the Early–Middle Jurassic transition. This turnover saw the demise of early neoichthyosaurians, rhomaleosaurid plesiosaurians and early plesiosauroids in favour of ophthalmosaurid ichthyosaurians and cryptoclidid and pliosaurid plesiosaurians, clades that will dominate the Late Jurassic and, for two of them, the entire Early Cretaceous as well. The fossil record of this turnover is however extremely poor and this change of dominance appears to be spread across the entire middle Toarcian–Bathonian interval. We describe a series of ichthyosaurian and plesiosaurian specimens from successive geological formations in Luxembourg and Belgium that detail the evolution of marine reptile assemblages across the Early–Middle Jurassic transition within a single area, the Belgo–Luxembourgian sub-basin. These fossils reveal the continuing dominance of large rhomaleosaurid plesiosaurians, microcleidid plesiosaurians and Temnodontosaurus-like ichthyosaurians up to the latest Toarcian, indicating that the structuration of the upper tier of Western Europe marine ecosystems remained essentially constant up to the very end of the Early Jurassic. These fossils also suddenly record ophthalmosaurid ichthyosaurians and cryptoclidid plesiosaurians by the early Bajocian. These results from a geographically-restricted area provide a clearer picture of the shape of the marine reptile turnover occurring at the early–Middle Jurassic transition. This event appears restricted to the sole Aalenian stage, reducing the uncertainty of its duration, at least for ichthyosaurians and plesiosaurians, to 4 instead of 14 million years.

Introduction

A series of diapsid clades dominated marine ecosystems during the entire Mesozoic (Motani, 2009; Benson, 2013; Pyenson, Kelley & Parham, 2014; Kelley & Pyenson, 2015). This long-term dominance was, however, pulsed by a series of extinctions and turnovers (Massare, 1987; Callaway & Massare, 1989; Bardet, 1994, 1995; Benson et al., 2010; Benson & Druckenmiller, 2014), the most-studied of which being those happening during the latest Triassic (Thorne, Ruta & Benton, 2011; Benson, Evans & Druckenmiller, 2012; Fischer et al., 2014c; Dick & Maxwell, 2015; Moon & Stubbs, 2020) and at the Jurassic-Cretaceous boundary (Tennant et al., 2017; Fischer et al., 2012, 2013; Benson & Druckenmiller, 2014; Young et al., 2014a; Tennant, Mannion & Upchurch, 2016; Zverkov et al., 2018).

The dynamics of marine reptile turnovers that took place outside of geological period boundaries are less well known, despite evident modifications of assemblages, such as during the early Late Cretaceous (Bakker, 1993; Bardet et al., 2008; Fischer et al., 2016, 2018). The composition of marine reptile communities clearly changed across the Early–Middle Jurassic transition (Massare, 1987; Maxwell, Fernández & Schoch, 2012; Vincent et al., 2013). On one hand, early neoichthyosaurian ichthyosaurians and microcleidid plesiosaurians seemingly met their demise and rhomaleosaurid plesiosaurians nearly went extinct (Maxwell, Fernández & Schoch, 2012; Fischer et al., 2013; Benson, Zverkov & Arkhangelsky, 2015). On the other hand, ophthalmosaurid ichthyosaurians, thalattosuchian crocodylomorphs, cryptoclidid plesiosauroids and pliosaurids radiated, dominating the Late Jurassic and beyond (Motani, 1999a; Cau & Fanti, 2011; Ketchum & Benson, 2011a; Fischer et al., 2013; Benson & Druckenmiller, 2014; Cau, 2014; Moon, 2017; Foffa, Young & Brusatte, 2018; Campos, Fernández & Herrera, 2020).

However, the fossil record of marine reptiles across the middle Toarcian–Callovian interval is extremely poor (Bardet, 1994; Benson et al., 2010; Benson & Butler, 2011; Cau & Fanti, 2011; Maxwell, Fernández & Schoch, 2012) (Fig. 1) and no region preserves relevant fossiliferous successions spanning this transition, as a result of a generalized regression reducing the volume of marine epicontinental ecosystems (Bardet, 1994). These biases blur both the tempo and the severity of this turnover. We tackle this issue by describing and analysing the composition of the late Toarcian-Bajocian ichthyosaurians and plesiosaurians of Luxembourg and Belgium (northeastern part of Paris Basin; Fig. 2; see Johnson et al. (2018) for an account on thalattosuchians), extending the spatiotemporal extent of several Jurassic clades and precising the severity of the biotic turnover at the Early–Middle Jurassic transition.

Figure 1 Number of collections of marine reptiles (Ichthyosauria, Plesiosauria, Thalattosuchia, Pleurosauria, Angolachelonia) per stage.

Data extracted from the paleobiology database on the 26th March 2020 (see Acknowledgements for the main contributors of these data). Silhouettes originate from phylopic.org, Licensed under CC BY 3.0 SA: Dakosaurus by Dmitry Bogdanov and T. Michael Keesey; Meyerasaurus, Eurhinosaurus, Temnodontosaurus, Plesiopterys and Ophthalmosaurus by Gareth Monger; Rhomaleosaurus and Stenopterygius by Scott Hartmann; Peloneustes by Nobu Tamura and T. Michael Keesey; Albertonectes by Frank Denota (CC 1.0).

Figure 2 Map of the fossiliferous localities investigated.

Materials and Methods

Upper Toarcian of Luxembourg: Thouarsense, Pseudoradiosa and Aalensis zones

Towards the end of the Early Jurassic, the area of present-day Luxembourg was located in the northeast Paris Basin within the northwestern Peri-Tethys Ocean (Pieńkowski et al., 2008; Schintgen & Förster, 2013). In a shallow, near-coastal sea between the former landmasses of the Rhenish Massif in the north and the Vosges and Black Forest in the south, a thick siliclastic succession was deposited during the late Toarcian, with clayey to silty sediments in the lower parts and silty to sandy, iron-grain rich sediments in the upper part following a general coarsening upwards trend (Siehl & Thein, 1989). The material described herein includes a single, semi-articulated post-cranial ichthyosaur skeleton from the so-called “Couches à Astarte voltzi” (Dittrich, 1993), a clay-rich siltstone corresponding to level 11 of Delsate & Weis (2010), dated to the late Toarcian Thouarsense Ammonite Zone of the Grandcourt Formation.

All the other late Toarcian marine reptile remains from Luxembourg described herein were found in the overlying Minette ironstone Formation (Lucius, 1945). Deposition of the Minette successions took place under the influence of strong tidal currents (Teyssen, 1984) during the upper Toarcian Pseudoradiosa and Aalensis ammonite Zones (Di Cencio & Weis, 2020). The Aalenian Opalinum ammonite zone has not yet been detected in the area and probably forms a local hiatus (Maubeuge, 1972; Guérin-Franiatte & Weis, 2010).

Upper Toarcian of Belgium: Dispansum–Pseudoradiosa–Aalensis zones

The Mont–Saint–Martin Formation crops out in southern Belgium and includes iron-rich beds forming a slightly diachronous equivalent of the Minette ironstones in Luxembourg. These ironstones containing marine reptiles were deposited in a shallow marine setting during the late Toarcian (Dispansum–Aaalensis ammonite zones; Boulvain et al., 2001). One fossil (IRSNB Vert-06455-0001) was found just outside the Belgian border, in the Meurthe-et-Moselle department, north-eastern France (Fig. 2).

Middle-upper Aalenian of Luxembourg: Murchisonae and Concavum zones

The Toarcian part of the Minette ironstones in Luxembourg is locally overlain by a few meters of iron-rich marly sandstones and thin beds with calcareous nodules deposited during the Aalenian Murchisonae to Concavum ammonite zones (Guérin-Franiatte & Weis, 2010; Sadki, Weis & Braun, 2020). These deposits form the onset of a transgressive succession with increasingly finer sedimentation leading to the so-called “marnes micacées”, a succession of poorly lithified silty claystones with thin layers of phosphatic nodules, dated to the lowermost Bajocian Discites ammonite zone (Guérin-Franiatte & Weis, 2010).

Lower Bajocian of Luxembourg: Humphriesianum zone

The youngest marine strata in Luxembourg correspond to a succession of calcarenites, marl-limestone alternations and coral limestones deposited on the northeast margin of the Burgundy carbonate platform at the end of the lower Bajocian. The marine reptile remains described herein were all recovered from the so-called “Marnes sableuses d’Audun-le-Tiche”, dated to the lower Bajocian Humphriesianum ammonite zone (Delsate et al., 2018; Popov, Delsate & Felten, 2019). An indeterminate thalattosuchian has been reported from these strata in Luxembourg (Johnson et al., 2018).

Marine reptile collections data

We extracted all the fossil collections of Jurassic (Hettangian–Tithonian) marine reptiles (selected clades: Ichthyosauria, Plesiosauria, Thalattosuchia, Pleurosauria and Angolachelonia) from the paleobiology database (https://paleobiodb.org) on the 26th March 2020. While the taxonomic record of Jurassic marine reptiles on the paleobiology database is not complete, the record we extracted samples the vast majority of the fossil localities of the Jurassic. This record is used as a proxy for sampling intensity for visualization purposes only; we binned these data per geological stage and generate a plot of the number of collections over time in R v3.6.2 (R Core Team, 2016), using the geoscale v2.0 package (Bell, 2015) (Fig. 1; see Supplemental Files for the dataset and the R script).

Comparative Descriptions

Thouarsense zone fauna, upper Toarcian, Luxembourg

ICHTHYOSAURIA De Blainville, 1835

PARVIPELVIA Motani, 1999

THUNNOSAURIA Motani, 1999

BARACROMIA Fischer et al., 2013

Baracromia indet.

A single specimen (MNHNL TM212) is recorded 2011, from the Thouarsense zone: it is a partial articulated ichthyosaur comprising fifty-two centra, several ribs and gastralia, a partial coracoid, and two partial forefins (Fig. 3; Tables 1 and 2), from the Dudelange locality. All the preserved pre-apical centra are markedly rounded and anteroposteriorly short, bearing close similarities with Jurassic thunnosaurians such as Stenopterygius and Ophthalmosaurus (Buchholtz, 2001; Massare et al., 2006). Posterior dorsal centra maintain clear bicipital rib articulations and all ribs have a 8-shaped cross-section, which also suggest that this specimen is a thunnosaurian (Sander, 2000; V. Fischer, 2011, personal observation on NHMUK 2003, R5465, R498), even though this feature is likely an ambiguous synapomorphy, being recorded in Early–Middle Triassic specimens as well (Schmitz, 2005; A. J. Roberts, 2020, personal communication). The posterior apical centra are strongly compressed laterally, suggesting the presence of a lunate tailfin; although is the angle of the tailbent is unknown. The coracoid (Figs. 3A and 3B) possesses a wide anterior notch, unlike in leptonectids (Von Huene, 1951; Maisch & Matzke, 2000; Lomax, 2016; Lomax, Evans & Carpenter, 2018). The presence of a posterolateral emargination, a trait that is typical of Ichthyosaurus, Protoichthyosaurus and Temnodontosaurus (Home, 1819; Godefroit, 1993a; Lomax & Massare, 2015; Lomax, Porro & Larkin, 2019) cannot be assessed because the coracoid is incomplete (Fig. 3A). The humerus (Figs. 3C–3F) is incompletely-preserved, lacking the capitulum. The dorsal trochanter is proximodistally short, as it does not extend to mid-shaft, while the deltopectoral crest is elongated and parallel to the long axis of the humerus. The small dorsal trochanter suggest that this taxon is not an ophthalmosaurid (Motani, 1999a). The humerus forms a prominent anterior process that possesses a conspicuous, anterodorsally facing facet. This facet exhibits a pitted bone texture which suggests the presence of a cartilaginous cap in vivo. While many Early Jurassic ichthyosaurians possess a mesiodistally-thick anterodistal expansion of the humerus (Temnodontosaurus, Leptonectidae, Hauffiopteryx, Stenopterygius (Johnson, 1979; McGowan & Milner, 1999; McGowan, 2003; McGowan & Motani, 2003; Maisch, 2008; Caine & Benton, 2011; Martin et al., 2012), this expansion forms a flat facet but not for articulation with an anterior accessory epipodial element. The rounded and notched anterior surface of the radius also confirms the absence of such an element. This combination of features rules out earliest neoichthyosaurians and ophthalmosaurids (Maisch & Matzke, 2000; McGowan & Motani, 2003) and has only been reported in the poorly known, ?late Toarcian-lower Bajocian taxon Dearcmhara shawcrossi (Brusatte et al., 2015), although the facet is smaller and deeper in D. shawcrossi. Epipodial and proximal elements closely resemble those of Stenopterygius spp. and Chacaicosaurus cayi (Johnson, 1979; Fernández, 1994; Maxwell, Fernández & Schoch, 2012), being polygonal and dorsoventrally thick, with no spatium interosseum. The radius and the radiale are notched as in many early neoichthyosaurians (Von Huene, 1922; McGowan, 1974; Maxwell, 2012; Lomax & Massare, 2016). The intermedium forms two distal facets unequal in size (Fig. 3D): a large, distally facing facet for articulation with distal carpal 3 and a smaller, posterodistally facing facet for articulation with distal carpal 4 (the ‘latipinnate’ condition, although the latipinnate-longipinnate dichotomy fails to capture what is essentially a trait with continuous spectrum; see Mazin (1982) and Motani (1999b) for discussions). The specimen MNHNL TM212 is regarded as a non-ophthalmosaurid baracromian, resembling Stenopterygius spp., Chacaicosaurus cayi, and Dearcmhara shawcrossi.

Figure 3 Thoursense Zone fauna, late Toarcian, Luxembourg.

Selected anatomy of specimen MNHNL TM212. (A and B) left coracoid in ventral (A) and anteroventral (B) views. (C, E and F) left humerus in anterodorsal (C), distal (E) and proximal (F) views. (D) Left forefin in ventral view.

Table 1 Centra dimensions (in millimeters).

Specimen	Clade	Age	Axial_zone	Height	Length	Width	
ULgPA35961	Ichthyosauria	Toarcian	dorsal	78	35	86	
IRSNB_R_436	Ichthyosauria	Toarcian	caudal	118	48	118	
IRSNB_ R_437	Ichthyosauria	Toarcian	caudal	62	22	60	
IRSNB_Vert-06462-0005	Ichthyosauria	Toarcian	cervical	63	24	64	
IRSNB_Vert-11312-00007	Plesiosauria	Toarcian	dorsal	67	44	81	
MNHNL_DOU378	Ichthyosauria	Toarcian	caudal	190	65	NA	
MNHNL_DOU978	Plesiosauria	Toarcian	cervical	41	55	53	
MNHNL_DOU944	Ichthyosauria	Toarcian	dorsal	82	36	NA	
MNHNL_DOU998	Ichthyosauria	Toarcian	sacral	90	31	NA	
MNHNL_DOU722	Plesiosauria	Toarcian	dorsal	50	37	60	
MNHNL_DOU723	Plesiosauria	Toarcian	dorsal	55	30	63	
MNHNL_DOU954	Plesiosauria	Toarcian	pectoral	63	42	73	
MNHNL_DOU724	Plesiosauria	Toarcian	sacral	39	40	57	
MNHNL_DOU369a	Ichthyosauria	Toarcian	dorsal	68	32	NA	
MNHNL_DOU369b	Ichthyosauria	Toarcian	caudal	61	35	NA	
MNHNL_DOU369c	Ichthyosauria	Toarcian	caudal	63	32	NA	
MNHNL_HU384	Plesiosauria	Aalenian	caudal	32	NA	NA	
MNHNL_BU157	Plesiosauria	Aalenian	cervical	67	62	71	
MNHNL_TM212	Ichthyosauria	Toarcian	cervical	62	32	56	
MNHNL_BM779	Ichthyosauria	Bajocian	caudal	91	39	NA	
MNHNL_BM758	Ichthyosauria	Bajocian	caudal	100	39	NA	
MNHNL_BM725a	Ichthyosauria	Bajocian	cervical	83	38	NA	
MNHNL_BM725b	Ichthyosauria	Bajocian	caudal	56	26	NA	
MNHNL_BM461	Ichthyosauria	Bajocian	cervical	72	30	NA	
MNHNL_BM392a	Ichthyosauria	Bajocian	caudal	72	30	NA	
MNHNL_BM392b	Ichthyosauria	Bajocian	caudal	77	31	NA	
MNHNL_BM392c	Ichthyosauria	Bajocian	caudal	72	29	NA	
MNHNL_BM392d	Ichthyosauria	Bajocian	caudal	75	32	NA	
MNHNL_BM779	Ichthyosauria	Bajocian	cervical	87	35	90	

Table 2 List of the material and taxonomic identifications.

Specimen	Clade	Stage	Zone	Element	Identification	Note	Figured	
MNHNL TMP212	Ichthyosauria	Toarcian	Thouarsense	Partial skeleton	Baracromia indet.	Non-ophthalmosaurid	Figure 3	
MNHNL DOU307	Plesiosauria	Toarcian	Pseudoradiosa	Humerus	Rhomaleosauridae indet.		Figures 4A–4E	
MNHNL KA109	Plesiosauria	Toarcian	Pseudoradiosa	Humerus	Rhomaleosauridae indet.		Figures 4F–4H	
ULgPA35961	Ichthyosauria	Toarcian	Dispansum to Aalensis	Dorsal centrum	Parvipelvia indet.		No	
IRSNB Vert-06455-0001	Ichthyosauria	Toarcian	Dispansum to Aalensis	Centrum	Ichthyosauria indet.		No	
IRSNB Vert-00000-00801	Ichthyosauria	Toarcian	Dispansum to Aalensis	Centrum	Ichthyosauria indet.		No	
IRSNB Vert-06659-0001	Ichthyosauria	Toarcian	Dispansum to Aalensis	Centrum	Ichthyosauria indet.		No	
IRSNB Vert-00000-00800	Ichthyosauria	Toarcian	Dispansum to Aalensis	Left surangular	Parvipelvia indet.		No	
IRSNB Vert-00000-00804	Ichthyosauria	Toarcian	Dispansum to Aalensis	Dorsal centrum	Parvipelvia indet.	Temnodontosaurus-like	No	
IRSNB Vert-06462-0005	Ichthyosauria	Toarcian	Dispansum to Aalensis	Cervical centrum	Parvipelvia indet.		No	
IRSNB R 436	Ichthyosauria	Toarcian	Dispansum to Aalensis	Caudal centrum	Parvipelvia indet.	Temnodontosaurus-like	Figures 5A and 5B	
IRSNB R 437	Ichthyosauria	Toarcian	Dispansum to Aalensis	Caudal centrum	Parvipelvia indet.	Possible thunnosaurian	Figures 5C and 5D	
IRSNB R 438	Ichthyosauria	Toarcian	Dispansum to Aalensis	Tooth	Ichthyosauria indet.		Figures 5E and 5F	
IRSNB R 439	Ichthyosauria	Toarcian	Dispansum to Aalensis	Left angular	Parvipelvia indet.		Figures 5G and 5H	
IRSNB R 440	Ichthyosauria	Toarcian	Dispansum to Aalensis	Right quadrate	Ichthyosauria indet.		Figure 5I–5K	
IRSNB Vert-11312-00007	Plesiosauria	Toarcian	Dispansum to Aalensis	Dorsal centrum	Plesiosauria indet.		No	
MNHNL DOU369	Ichthyosauria	Toarcian	Aalensis	Centra	Parvipelvia indet.	Probable non-thunnosaurian	No	
MNHNL DOU353	Ichthyosauria	Toarcian	Aalensis	Coracoid	Parvipelvia indet.	Non-baracromian	Figures 6A–6C	
MNHNL DOU378	Ichthyosauria	Toarcian	Aalensis	Posterior dorsal to anterior caudal centrum	Parvipelvia indet.	Probable non-thunnosaurian	Figures 6D–6F	
MNHNL DOU998	Ichthyosauria	Toarcian	Aalensis	Caudal centrum	Parvipelvia indet.		Figure 6G	
MNHNL DOU944	Ichthyosauria	Toarcian	Aalensis	Dorsal centrum	Parvipelvia indet.		Figures 6H and 6I	
MNHNL DOU906	Plesiosauria	Toarcian	Aalensis	Tooth crown	Plesiosauria indet.		Figure 7A	
MNHNL DOU558	Plesiosauria	Toarcian	Aalensis	Humerus	Rhomaleosauridae indet.		Figure 7B	
MNHNL DOU324A	Plesiosauria	Toarcian	Aalensis	Humerus	Rhomaleosauridae indet.		Figures 7C–7F	
MNHNL DOU324B	Plesiosauria	Toarcian	Aalensis	Humerus	Rhomaleosauridae indet.	Juvenile	Figure 7G	
MNHNL DOU954	Plesiosauria	Toarcian	Aalensis	Posterior pectoral/anterior dorsal	Rhomaleosauridae indet.		Figures 8A and 8B	
MNHNL DOU978	Plesiosauria	Toarcian	Aalensis	Cervical centrum	cf. Microcleidus		Figures 8C–8G	
MNHNL DOU723	Plesiosauria	Toarcian	Aalensis	Pectoral vertebra	Rhomaleosauridae indet.		Figures 8H–8J	
MNHNL DOU724	Plesiosauria	Toarcian	Aalensis	Sacral vertebra	Plesiosauria indet.		Figures 8K–8N	
MNHNL DOU722	Plesiosauria	Toarcian	Aalensis	Pectoral vertebra	Rhomaleosauridae indet.		Figures 8O–8Q	
MNHNL BU157	Plesiosauria	Aalenian	Concavum or Murchisonae	Cervical vertebra	Plesiosauroidae indet.	Non-cryptoclidid	Figures 9A–9D	
MNHNL HU384	Plesiosauria	Aalenian	Murchisonae	Caudal centrum	Plesiosauria indet.	Juvenile	Figures 9E–9G	
MNHNL HU242	Ichthyosauria	Aalenian	Concavum	Partial rostrum	Ichthyosauria indet.		Figures 9H and 9I	
MNHNL BM360	Ichthyosauria	Bajocian	Humphriesianum	Partial neural spine and ribs	Ichthyosauria indet.		No	
MNHNL BM758	Ichthyosauria	Bajocian	Humphriesianum	Caudal centrum	Parvipelvia indet.		No	
MNHNL BM725	Ichthyosauria	Bajocian	Humphriesianum	Cervical and caudal centra	Parvipelvia indet.		No	
MNHNL BM461	Ichthyosauria	Bajocian	Humphriesianum	Cervical centrum	Parvipelvia indet.		No	
MNHNL BM782	Plesiosauria	Bajocian	Humphriesianum	Propodial	Cryptoclididae indet.		Figures 10A–10F	
MNHNL BM392	Ichthyosauria	Bajocian	Humphriesianum	Five caudal centrum and one sclerotic plate	Parvipelvia indet.		Figures 10G and 10H	
MNHNL BM780_781	Ichthyosauria	Bajocian	Humphriesianum	Surangular, teeth	Ophthalmosauridae indet.		Figures 10I–10O	
MNHNL BM779	Ichthyosauria	Bajocian	Humphriesianum	Partial skeleton	Ophthalmosauridae indet.		Figure 11	

Pseudoradiosa zone fauna, upper Toarcian, Luxembourg

PLESIOSAURIA De Blainville, 1835

Rhomaleosauridae Nopcsa, 1928

Rhomaleosauridae indet.

Two large plesiosaurian propodials have been recovered from Pseudoradiosa zone: one (MNHNL DOU307) from the “Couche grise” of the Rodange locality, and one (MNHNL KA109) from the “Couche noire” of the Esch-sur-Alzette locality (Table 2).

MNHNL DOU307 is a complete left propodial (proximal-distal length: 390 mm; distal width: 210 mm) with a weak posterior and dorsal curvature (Figs. 4A–4E). The shaft is as wide as the capitulum, and there is no anterior expansion indicating that this propodial is an early plesiosaurian humerus (Storrs, 1997; Bardet, Godefroit & Sciau, 1999; O’Keefe, 2004; Smith & Vincent, 2010; Benson, Evans & Druckenmiller, 2012; Vincent & Storrs, 2019). The dorsal tuberosity is as anteroposteriorly wide as the capitulum and is weakly demarcated. A bulge with muscle scars is present on the ventral surface of the shaft, 30 mm distally to the capitulum. A very short prexial flange is present distally, forming a small, anterodistally facing triangular facet. The radial/tibial facet is flat and marks an angle of ca. 45° with the ulnar/fibular facet. This facet is convex and rugose, indicating the presence of an extensive cartilage layer. A larger postaxial flange is present, forming distally a semioval facet that is hardly discernible from the ulnar/fibular facet. There is no distal ridge, unlike in the cryptoclidid Colymbosaurus megadeirus (Benson & Bowdler, 2014; Roberts et al., 2017). The absence of a marked curvature (the anterior surface is straight in dorsal and ventral views) and the presence of a conspicuous postaxial flange suggest rhomaleosaurid affinities (Smith & Vincent, 2010; Smith & Benson, 2014; Smith, 2015) (with the exception of Lindwurmia thiuda Vincent & Storrs, 2019), which are reinforced by the large size of the propodial.

Figure 4 Pseudoradiosa Zone fauna, late Toarcian, Luxembourg.

(A–E) rhomaleosaurid left humerus MNHNL DOU307 in ventral (A), anterior (B), dorsal (C), proximal (D), and distal (E) views. (F–H) rhomaleosaurid humerus MNHNL KA109 in dorsal/ventral (F), proximal (H) and distal (H) views.

MNHNL KA109 is the distal end of a plesiosaurian propodial; the anterior-posterior asymmetry suggests it is also a humerus (Figs. 4F–4H). It resembles MNHNL DOU307 in lacking a preaxial flange and the postaxial flange is smaller than in MNHNL DOU307. The distal surface is markedly rounded in dorsoventral view. A prominent, strongly pitted anteroposterior ridge textures of the distal surface, obliterating radial/tibial and ulnar/fibular facets. The postaxial flange forms a triangular and concave facet posterior to the distal ridge. An anteroposteriorly elongated distal ridge is regarded as an autapomorphic feature of Colymbosaurus megadeirus (Benson & Bowdler, 2014; Roberts et al., 2017; Arkhangelsky et al., 2019) but the lack of an extensive postaxial flange precludes such a referral; the size of the humerus and its similarities with MNHNL DOU307 suggest rhomaleosaurid affinities.

Dispansum–Pseudoradiosa–Aalensis zones, upper Toarcian, Belgium

ICHTHYOSAURIA De Blainville, 1835

Ichthyosauria indet.

IRSNB Vert-06455-0001, IRSNB Vert-00000-00801 and IRSNB Vert-06659-0001 are ichthyosaurian centra, heavily eroded and damaged. Because their original shapes cannot be asserted, we refrained from calculating their shape ratios and assign them to Ichthyosauria indet (Table 2).

IRSNB R 438 (Figs. 5E and 5F) is a distal tooth that preserves the crown and the acellular cementum ring. The crown is recurved and robust (height: 14.83 mm, basal diameter: 8.14 mm, giving a crown shape ratio of 1.82), as is usually the case for distal ichthyosaurian teeth (Motani, 1997; McGowan & Motani, 2003). Apicobasal ridges texture the crown, unlike in leptonectids (Maisch, 1998; Maisch & Matzke, 2003; McGowan & Motani, 2003; Lomax, 2016) and the acellular cementum is itself ridged. This combination of features match that of Ichthyosaurus, Protoichthyosaurus and Temnodontosaurus (McGowan, 1973; Godefroit, 1993a; Vincent et al., 2014; Brusatte et al., 2015; Lomax, Porro & Larkin, 2019). But it is also found in more primitive taxa such as Contectopalatus and Cymbospondylus (Maisch & Matzke, 2001; Klein et al., 2020). As a result, this specimen is referred to as Ichthyosauria indet. (Table 2).

Figure 5 Ichthyosaurians of the late Toarcian of Luxembourg and Belgium.

(A and B) large, Temnodontosaurus-like (Parvipelvia indet.) caudal centrum IRSNB R 436 in anterior (A) and lateral (B) views. (C and D) probable thunnosaurian caudal centrum IRSNB R 437 in anterior (C) and dorsal (D) views. (E and F) ichthyosaurian distal tooth IRSNB R 438 in labial (E) and mesial (F) views. (G and H) probable non-ophthalmosaurid (Parvipelvia indet.) left angular IRSNB R 439 in lateral (G) and medial (H) views. (I–K) ichthyosaurian right quadrate IRSNB R 440 in lateral (I), medial (J) and posterior (K) views.

IRSNB R 440 (Figs. 5I–5K) is a partial ichthyosaurian right quadrate, preserving the condyle, the stapedial facet and the posteroventral part of the anterior lamella. The condyle is massive, oval, but appears moderately short anteroposteriorly. The medial and lateral surfaces are pitted by numerous small foramina; the stapedial facet is semi-circular thickened ventral and posterior edges. This quadrate has the same size (maximal condylar width: 71 mm) and morphology as that of Temnodontosaurus platyodon (Godefroit, 1993a). In the absence of additional evidence, this quadrate is referred to Ichthyosauria indet (Table 2) but indicates the presence of large ichthyosaurians by the latest Toarcian.

PARVIPELVIA Motani, 1999

Parvipelvia indet.

ULg-PA 35.961 is an ichthyosaurian anterior dorsal centrum with a damaged dorsal surface. Its height-length ratio is 78 mm/35 mm = 2.23 (Table 1). The centrum is rounded as in parvipelvians (Merriam, 1908; Maisch & Matzke, 2000) and the diapophyses and parapophyses strongly protrude from the body of the centrum, as often seen in neoichthyosaurians (Owen, 1865; Von Huene, 1922).

IRSNB Vert-00000-00800 is a partial left ichthyosaurian surangular, probably associated with the angular IRSNB R 439. It consists of posterior third, where the angular in a mesiolaterally-flattened bone with a slightly convex lateral surface and a slightly concave medial surface for accommodation of the Meckelian canal. The dorsal surface of the surangular is saddle-shaped, forming a long and shallow concavity anterior to the coronoid process. The preserved morphology of IRSNB Vert-00000-00800 is not diagnostic, but we assign it to Parvipelvia indet. because of its probable association with the angular IRSNB R 439 (see below).

IRSNB R 439 (Figs. 5G and 5H) is a partial left ichthyosaurian angular, bearing the typical double-grooved dorsal surface. The lateral wall of the angular is dorsoventrally short anteriorly, and only increases in size posterior to the level of the coronoid process. This narrow lateral exposure of the angular suggests that this specimen does not belong to Ophthalmosauridae, as this clade is notably characterized by large exposure of the angular in lateral view (Motani, 1999a).

IRSNB R 436 (Figs. 5A and 5B) and IRSNB Vert-00000-00804 are large ichthyosaurian centra. IRSNB R 436 is 118 mm high and 48 mm long, giving a height-length ratio of 2.45 (Table 1).; IRSNB Vert-00000-00804 is damaged on the edges and will not be measured but is of comparable size. The lateral surfaces of these centra are rounded in anteroposterior view, as in parvipelvians (Merriam, 1908; Maisch & Matzke, 2000), but their size departs from that of Early Jurassic thunnosaurians, even particularly large individuals (Maxwell, 2012; Lomax & Sachs, 2017). These centra rather match the size and shape of the dorsal-caudal centra of some large non-thunnosaurian neoichthyosaurians such as Temnodontosaurus spp. (Godefroit, 1993a; Martin et al., 2012) and Excalibosaurus costini (McGowan, 2003). IRSNB R 436 possesses a single apophysis (the parapophysis) placed laterally, suggesting it is middle preflexural caudal (Von Huene, 1922; McGowan & Motani, 2003). IRSNB Vert-00000-00804 possesses a diapophysis and a parapophysis placed ventrolaterally and is regarded as a posterior dorsal centrum. The size and shape of IRSNB R 436 and IRSNB Vert-00000-00804 match those of large early neoichthyosaurians and but are regarded as Parvipelvia indet. in the absence of unambiguous synapomorphies (Table 2).

IRSNB R 437 (Figs. 5C and 5D) is an ichthyosaurian anterior caudal centrum. Its height-length ratio is 65 mm/22 mm = 2.94 (Table 1).; this high shape ratio close to 3 might suggest thunnosaurian affinities (Buchholtz, 2001). The lateral surface of the centrum is rounded as in parvipelvians (Merriam, 1908; Maisch & Matzke, 2000) and slightly oval, as the ventral surface is flattened. The parapophyses are located lateroventrally.

IRSNB Vert-06462-0005 is an ichthyosaurian cervical centrum. Its height-length ratio is 63 mm/24 mm = 2.62. The lateral surface of the centrum is rounded as in parvipelvians (Merriam, 1908; Maisch & Matzke, 2000). A faint ventral keel is present, giving the centrum a pentagonal shape. Such a shape is often seen in the cervical/anterior dorsals of parvipelvians (Von Huene, 1922; Fischer et al., 2014b; Lomax, Porro & Larkin, 2019) and is not of diagnostic value at a lower taxonomic level at the current state of our knowledge.

PLESIOSAURIA De Blainville, 1835

Plesiosauria indet.

IRSNB Vert-11312-00007 is a plesiosaurian dorsal centrum, as evidenced by the absence of transverse processes. Its height-length ratio is 67 mm/44 mm = 1.52. The centrum is poorly preserved and we assign it to Plesiosauria indet. (Table 2).

Aalensis zone fauna, upper Toarcian, Luxembourg

The material from the Aalensis zone of the Minette ironstones is more abundant. We identified 12 ichthyosaurians and 8 plesiosaurians in the MNHL collections. Most of the material consist of isolated centra, complemented by three plesiosaurian propodials, one ichthyosaur coracoid, and one plesiosaurian tooth crown.

ICHTHYOSAURIA De Blainville, 1835

PARVIPELVIA Motani, 1999

Parvipelvia indet.

MNHNL DOU944 (Figs. 6H and 6I) is an ichthyosaurian anterior dorsal centrum from an unknown locality. A central keel is not present but the centrum forms a ventral expansion, giving the centrum a slight pentagonal shape. The diapophysis is small and rounded, as in parvipelvians and unlike in more primitive forms (Merriam, 1908; Von Huene, 1922) and is strongly protruding and has a rounded distal surface. Shallow dorsoventral ridges connect to the base of the diapophysis, anterior and posteriorly. The parapophysis is flatter and less prominent and its anterior margin merges with the anterior edge of the centrum. Such a pentagonal shape is often seen in parvipelvians (Von Huene, 1922; Fischer et al., 2014b; Lomax, Porro & Larkin, 2019) and is not of diagnostic value at a lower taxonomic level at the current state of our knowledge.

Figure 6 Ichthyosaurians of the Aalensis Zone, late Toarcian, Luxembourg.

(A–C) left coracoid of a non-baracromian parvipelvian MNHNL DOU353 in ventral (A), lateral (B), and posterior (C) views. (D–F) large parvipelvian caudal centrum MNHNL DOU378 in dorsal (D), anterior (E), and cross-sectional (F) views. (G) small parvipelvian caudal centrum MNHNL DOU998 in anteroventral view. (H and I) parvipelvian dorsal centrum MNHNL DOU944 in anterior (H) and lateral (I) views.

MNHNL DOU998 (Fig. 6G) is an anterior caudal centrum from an unknown locality. The centrum has a rounded outline, as in usually parvipelvian ichthyosaurians (Merriam, 1908; e.g., Maisch & Matzke, 2000; McGowan & Motani, 2003). Its height/length ratio is 90/31 = 2.9. This high ratio close to 3 might suggest thunnosaurian affinities (Buchholtz, 2001), but in the absence of additional data, this specimen is regarded as Parvipelvia indet. (Tables 1 and 2).

MNHNL DOU369 is a series of disarticulated ichthyosaur centra. One originates from the mid-thoracic region and two others from the anterior caudal region, their respective height/length ratios are as follows: 68/32 = 2.1, 61/35 = 1.74 and 63/32 = 1.96. The mid-thoracic resembles MNHNL DOU944. The rather elongated anterior caudal suggest these centra do not belong to a thunnosaurian, but rather to an early parvipelvian (Buchholtz, 2001).

MNHNL DOU353 (Figs. 6A–6C) is a fragmentary left coracoid from an unknown locality. It preserves the lateral part and a fragment of the intercoracoid facet. The ventral surface is moderately saddle-shaped (intercoracoid facet is only 56 mm thick while the mesiolateral length is 180 mm). The glenoid facet is large (anteroposterior length = 75 mm) and oval, while the scapular facet forms a small triangle (anteroposterior length = 39 mm) and forms a ca. 60° angle with the glenoid facet. The posterolateral emargination is wide and deep: the coracoid surface posterior to the glenoid faces posteriorly and not posterolaterally, similar to the condition seen in Ichthyosaurus communis, Ichthyosaurus anningae, Protoichthyosaurus postaxialis and many but not all specimens of Temnodontosaurus platyodon (Home, 1819; Godefroit, 1993a; Lomax & Massare, 2015; Lomax, Porro & Larkin, 2019). The Triassic shastasaurid Shatasaurus neoscapularis also possesses a wide posterior notch, but the coracoid of that taxon is different, having a very wide anterior notch (McGowan, 1994). We regard this coracoid as belonging to a non-baracromian parvipelvian.

MNHNL DOU378 (Figs. 6D–6F) is a very large centrum from an unknown locality. It originates from the posterior dorsal–sacral–anterior caudal region (length: 65 mm and diameter: 175 mm, estimated height: 180–185 mm, H/L ratio: 2.79–2.84; Table 1). Its size departs from the Early Jurassic thunnosaurians (Ichthyosaurus and Stenopterygius, which have much smaller centra even particularly large individuals (Maxwell, 2012; Lomax & Sachs, 2017)). It rather matches the size and shape of the anteriormost caudal centra of some large non-thunnosaurian neoichthyosaurians such as Temnodontosaurus spp. (Godefroit, 1993a; Martin et al., 2012) and Excalibosaurus costini (McGowan, 2003) (the anterior caudals are much longer in Eurhinosaurus longirostris (V. Fischer, 2008, personal observation on MNHN 1946-20).

PLESIOSAURIA De Blainville, 1835

Plesiosauria indet.

MNHNL DOU906 (Fig. 7A) is a plesiosaurian tooth crown from the Esch-sur-Alzette locality. It is slender (length/basal diameter = 23/8 = 2.875), labiolingually flattened and recurved indicating it does not belong to derived rhomaleosaurids and thalassophoneans (Owen, 1865; Smith & Vincent, 2010; Ketchum & Benson, 2011b; Smith & Benson, 2014). The crown bears fine striations along the entire surface.

Figure 7 Plesiosaurians of the Aalensis Zone, late Toarcian, Luxembourg.

(A) plesiosaurian tooth crown MNHNL DOU906 in ?mesial view. (B) rhomaleosaurid right humerus MNHNL DOU558 in dorsal view. (C–F) rhomaleosaurid right humerus MNHNL DOU324a in proximal (C), dorsal (D), posterior (E) and distal (F) views. (G) juvenile rhomaleosaurid propodial MNHNL DOU324b.

MNHNL DOU724 (Figs. 8K–8N) is a plesiosaurian sacral vertebra probably originating from the Esch-sur-Alzette locality. The articular surface is strongly dorsoventral depressed and the centrum is as wide as long. As in MNHNL DOU722, MNHNL DOU723 and MNHNL DOU954, the subcentral foramina are widely spaced and are separated by a well-rounded/convex surface. Similar to the other centra, the dorsal edge of the articular surface is notched by the neural canal. The transverse process is elongated, stout, and points posterolaterally. The rib facet is 8-shaped (waisted), this time with a larger, ear-shaped dorsal part and a smaller, semioval ventral part. The neural canal is also 8-shaped, resembling a ‘plesiosauroid’ pectoral vertebra from the Aalenian-Bajocian of Australia (Kear, 2012). Despite having similar size to the vertebrae we referred to as Rhomaleosauridae indet., none of the feature we highlight above are recorded in Rhomaleosaurus (Smith, 2013; Smith & Benson, 2014) and their representation in other derived rhomaleosaurids is unclear. Accordingly, we regard this specimen as Plesiosauria indet.

Figure 8 Plesiosaurian centra of the Aalensis Zone, late Toarcian, Luxembourg.

(A and B) rhomaleosaurid pectoral vertebra MNHNL DOU954 in anterior (A) and ventral (B) views. (C–G) cf. Microcleidus cervical vertebra MNHNL DOU978 in anterior (C), ventral (D), dorsal (E), lateral (F) and oblique (G) views. (H–J) rhomaleosaurid pectoral vertebra MNHNL DOU723 in anterior (H), lateral (I) and posterior (J) views. (K–N) plesiosaurian sacral vertebra MNHNL DOU724 in dorsal (K), anterior (L), posterolateral (M) and ventral (N) views. (O–Q) rhomaleosaurid pectoral vertebra MNHNL DOU722 in anterior (O), lateral (P) and posterior (Q) views.

Rhomaleosauridae Nopcsa, 1928

Rhomaleosauridae indet.

MNHNL DOU722 (Figs. 8O–8Q) and MNHNL DOU723 (Figs. 8H–8J) are two plesiosaurian pectoral vertebrae from the Esch-sur-Alzette locality. The articular surface of the centrum is oval, with a dorsal notch corresponding to the neural canal. The subcentral foramina are displaced laterally, being located on the ventrolateral surface. The ventral surface in between the subcentral foramina is clearly transversely convex. The rib facets are oval, connect to the neural arch, and their long axis is vertical; they extend ventrally below the level of the notochordal pit. The semi-circular prezygaphophyses face dorsomedially and are clearly separated from one another; their ventral margin is set below the centre of the neural canal, even though the neural canal is large. Anteriorly, the neural spine forms a basal triangular cavity containing a paired ridge. The height of this trough is about 1/3 of the total height of the neural spine. On the posterior surface, the basal region of the neural spines forms a deep tear-shaped concavity that extends for about one half of the total height of the neural spine, and appears similar to the condition of Microcleididae (Bardet, Godefroit & Sciau, 1999; Schwermann & Sander, 2011) and Rhomaleosaurus (Smith & Benson, 2014). However, the neural spines are much shorter than in microclidids (Owen, 1865; Bardet, Godefroit & Sciau, 1999). The neural spine possesses a posterior ridge while its anterior surface is concave, as in Rhomaleosaurus thortoni (Smith & Benson, 2014). The neural spine thickens dorsally and its dorsal half is posteriorly inclined, unlike in Westphaliasaurus simonensii (Schwermann & Sander, 2011). We assign these vertebrae to Rhomaleosauridae indet. (Table 2).

MNHNL DOU954 (Figs. 8A and 8B) is a posterior pectoral/anterior dorsal vertebra from an unknown locality. It is essentially similar to MNHNL DOU722 and MNHNL DOU723. The main difference is the existence of ventral expansion of the centrum, even if a clear ventral keel remains absent. This ventral expansion gives the articular surface of the centrum a heart shape. The neural spine is concave anteriorly and bears a median ridges similar to the condition of Rhomaleosaurus thorntoni (Smith & Benson, 2014). We assign this vertebra to Rhomaleosauridae indet. (Table 2).

MNHNL DOU558 (Fig. 7B) from the Belvaux locality and MNHNL DOU324a (Figs. 7C–7F) and MNHNL DOU324b (broken, juvenile; Fig. 7G) from the Esch-sur-Alzette locality are three plesiosaurian humeri. These strongly resemble MNHNL DOU307: the shaft is long and slightly curved posteriorly and dorsally, the dorsal tuberosity is wide, a boss with muscle scars is present ventrally to the capitulum, the preaxial lamella is very short and forms a small triangular distal facet, the postaxial lamella is longer and forms a semioval posterodistal facet, radial/tibial and ulnar/fibular facets form a 45–55° angle and no distal ridge is present. The only difference is the position of the dorsal tuberosity, which is offset ventrally, creating a flattened, dorsomedially-facing surface between the dorsal part of the capitulum and the ventral part of the dorsal tuberosity. The larger propodial (MNHNL DOU324A) measures 331 mm in proximal distal length and 168 mm is maximal distal width. The smaller ‘adult’ propodial (MNHNL DOU558) measures 309 mm in proximodistal length and 150 mm in distal width.

Microcleididae Benson et al. 2012

Microcleidus Watson, 1909

cf. Microcleidus

MNHNL DOU978 (Figs. 8C–8G) is a plesiosaurian cervical centrum from an unknown locality. The centrum is slightly wider than high and is markedly elongated (height = 42 mm, width = 53 mm, length = 55 mm) unlike in derived rhomaleosaurids and thalassophoneans, which have anteroposteriorly short cervical centra (Owen, 1865; Smith & Vincent, 2010; Ketchum & Benson, 2011b; Smith & Benson, 2014). The ventral surface is flattened as in Microcleidus (Owen, 1865; Vincent et al., 2017) and bears two subcentral foramina. There is no ventral keel, unlike in rhomaleosaurids (O’Keefe, 2001a; Smith & Dyke, 2008; Benson & Druckenmiller, 2014). The lateral surface bears a (very) faint anteroposterior ridge; a more conspicuous lateral ridge is present in Microcleidus spp. (Owen, 1865; Bardet, Godefroit & Sciau, 1999; Vincent et al., 2017). The neural arch is fused with the centrum. The orientation of bone fibers on the right side suggests that the suture is V-shaped, which is mainly seen in rhomaleosaurids (Benson, Evans & Druckenmiller, 2012) and in the early pliosaurid Hauffiosaurus (Benson et al., 2011). The neural canal is very small in diameter, resulting in zygapophyses that are located close to the centrum, as in Microcleidus (Owen, 1865; Bardet, Godefroit & Sciau, 1999) and unlike in Plesiosaurus dolichodeirus (Owen, 1865), Westphaliasaurus simonensii (Schwermann & Sander, 2011), Hauffiosaurus spp. (White, 1940; Benson et al., 2011) The prezygapophyses are clearly separated, medially-inclined and their ventral surface is located more ventrally than the centre of the neural canal. The rib facets is 8-shaped, being formed of two semi-ovals separated by an anterior and a posterior furrows, as in Plesiosaurus dolichodeirus (Owen, 1865; Storrs, 1997) and unlike in many microcleidids (Owen, 1865; Schwermann & Sander, 2011), Hauffiosaurus (White, 1940) and cryptoclidids (Andrews, 1910a; Brown, 1981; Knutsen, Druckenmiller & Hurum, 2012a, 2012b). The rib facet in posterior cervical centra is separated by a groove Microcleidus melusinae (Vincent et al., 2017). This centrum thus bears strong similarities with Microcleidus and is regarded as cf. Microcleidus (Table 2).

Upper Aalenian fauna

ICHTHYOSAURIA De Blainville, 1835

Ichthyosauria indet.

MNHNL HU242 (Concavum zone; Figs. 9H and 9I) is a small partial ichthyosaur rostrum from te Rumelange locality. It contains the anterior part of the left dentary (or right premaxilla), showing the labial and lingual walls. Two teeth are preserved in situ. The root and the acellular cementum ring have a circular cross-section and are perfectly smooth externally, unlike in Ichthyosaurus, Protoichthyosaurus, and Temnodontosaurus (McGowan, 1973; Godefroit, 1993a; Vincent et al., 2014; Brusatte et al., 2015; Lomax, Porro & Larkin, 2019) and there is no evidence for plicidentine (unlike other Jurassic ichthyosaurians for which this feature is known (Maxwell, Caldwell & Lamoureux, 2012)), but this latter feature cannot be used for a taxonomic purpose at the present state of our knowledge. The teeth are of a maximal diameter of 8 mm and estimated 20 mm apicobasal length; the size and shape of the tooth fits within the range of early baracromians (Godefroit, 1993b, 1994a; Fernández, 1994; Maisch, 2008; Maxwell, Fernández & Schoch, 2012) but in the lack of additional evidence, this material is here referred to Ichthyosauria indet (Table 2).

Figure 9 Late Aalenian fauna, Luxembourg.

(A–D) non-cryptoclidid plesiosauroid cervical centrum MNHNL BU157 in anterior (A), dorsal (B), ventral (C) and oblique (D) views. (E–G) plesiosaur juvenile caudal centrum MNHNL HU384 in lateral (E), anterior (F) and anteroventral (G) views. (H and I) Fragmentary ichthyosaurian rostrum MNHN HU242.

PLESIOSAURIA De Blainville, 1835

Plesiosauria indet.

MNHNL HU384 (Murchisonae zone; Figs. 9E–9G) is a small caudal centrum of a plesiosaurian of a juvenile individual, originating from the Tétange locality. Its height/width ratio is 32/36 mm = 0.88. Two small triangular chevron facets are present ventrally, contacting the edge of the articular surface. The ventral surface is flat. A rounded rib facet is present on the lateral surface. The neural arc is disconnected from the centrum and absent; the facets appear diamond shaped. Caudal vertebrae usually have less diagnostic features than other vertebrae in plesiosaurians, as could be inferred from their respective number of phylogenetic characters (Benson & Druckenmiller, 2014); this, coupled to the immaturity of the specimen would make an assignment ambiguous and we refer this material to Plesiosauria indet (Table 2).

PLESIOSAUROIDEA Gray, 1825

Plesiosauroidea indet.

MNHNL BU157 (Concavum or Murchisonae zone; Figs. 9A–9D) is a moderately elongated plesiosaurian cervical vertebra from the Rumelange locality. The shape of the centrum suggestes it does not belong to derived rhomaleosaurids and thalassophoneans, which have anteroposteriorly short centra (Owen, 1865; Smith & Vincent, 2010; Ketchum & Benson, 2011b; Smith & Benson, 2014). The ventral surface is flattened and bears two subcentral foramina. No ventral keel is present. The edge of the centrum is rugose, as sometimes seen in various plesiosaurian clades (Owen, 1865; Seeley, 1874a; Fischer et al., 2020). An anteroposteriorly elongated bulge (rather than a ridge) is present on the dorsolateral surface. This bulge is separated from the rib facet by a median concave area. The rib facet is roughly oval but is very strongly waisted by an anterior and a posterior notch, giving the facet a marked eight shape; this structure recalls—but is more marked than in—the late Toarcian centra MNHNL DOU978. The neural arch is fully fused to the centrum and no suture is discernible. A pair of supracentral foramina is present on the floor of foramen magnum. While MNHNL DOU978 could clearly be attributed to Microcleidus, this vertebra—although superficially similar—is less elongated, possesses a lateral bulge rather than a ridge, and does not preserve the zygapophyses. As a result, a referral to Microclididae is too ambiguous. Nevertheless, the presence of an 8-shaped rib facet precludes a referral to cryptoclidids (Andrews, 1910a; Brown, 1981; Knutsen, Druckenmiller & Hurum, 2012a, 2012b). Accordingly, this specimen is referred to as a non-cryptoclidid plesiosauroid (Table 2).

Lower Bajocian fauna

ICHTHYOSAURIA De Blainville, 1835

Ichthyosauria indet.

MNHNL BM360 from the Rumelange locality contains one partial neural spine and two proximal parts of bicipital ribs. The neural spine bears a dorsoventrally-oriented median ridge on its anterior surface. The taxonomic values of these features is low and we assign this material to Ichthyosauria indet. (Table 2).

PARVIPELVIA Motani, 1999

Parvipelvia indet.

MNHNL BM758 is an ichthyosaurian caudal centrum from the Rumelange locality. Its shape is rounded, with a slightly flattened ventral surface, as in Parvipelvia (Merriam, 1908; Maisch & Matzke, 2000). Its height/length ratio is 100 mm/38 mm = 2.63.

MNHNL BM725 from the Rumelange locality contains a cervical and a caudal ichthyosaurian centra. The cervical centrum has a height/length ratio of 83 mm/38 mm=2.18 while the same ratio for the caudal centrum is 56 mm/26 mm=2.15. The centra are rounded in shape and exhibit small and rounded apophyses, as in parvipelvians and unlike in more primitive forms (Merriam, 1908; Von Huene, 1922).

MNHNL BM461 is an ichthyosaurian cervical centrum from the Rumelange locality. Its height/length ratio is 72 mm/30 mm = 2.4. The centrum is rounded in shape and exhibits small and rounded apophyses, as in parvipelvians and unlike in more primitive forms (Merriam, 1908; Von Huene, 1922).

MNHNL BM392 (Figs. 10G and 10H) from the Rumelange locality contains five caudal centra and one sclerotic plate of a single ichthyosaurian specimen. The height/length ratios of the first four caudal centra are as follows: 72 mm/30 mm = 2.4, 77 mm/31 mm = 2.48, 72 mm/29 mm = 2.48, 75 mm/32 mm = 2.34 (Table 1). Their rounded shape indicates they belong to Parvipelvia (Merriam, 1908; Von Huene, 1922; Maisch & Matzke, 2000) (Table 2). The sclerotic plate bears radiating striations on its lateral surface and its internal and external edges are crenulated, as is usually the case in ichthyosaurians (Andrews, 1910a; McGowan, 1973; Fischer et al., 2014a). The external third of the plate is deflected, facing dorsally.

Figure 10 Humphresianum Zone fauna, early Bajocian, Luxembourg.

(A–F) cryptoclidid propodial MNHNL BM782 in anterior (A), proximal (B), dorsal (C), ventral (D), posterior (E) and distal (F) views. (G and H) parvipelvian MNHNL BM392 centra (G) and sclerotic element in lateral view (H). (I–O) ophthalmosaurid surangular and teeth MNHNL BM780_781: anterior tooth in labial (I) and basal (J) views; mid-rostrum tooth in labial (K) and basal (L) views; posterior tooth in labial view (M); right angular in lateral (N) and medial (O) views. Abbreviation: PAE, posterior accessory epipodial element.

OPHTHALMOSAURIDAE Baur, 1897

Ophthalmosauridae indet.

MNHNL BM780–BM781 (Figs. 10I–10O) is a nearly complete right ichthyosaurian surangular (MNHNL BM780), associated with seven teeth (bearing the collection number MNHNL BM781), originating from the Rumelange locality. The surangular is straight in all planes (only its posterior quarter is slightly deflected medially) and bears a thickened dorsal margin, giving it a tear-shaped cross-section. A lateral, anteroposteriorly elongated and posteriorly-deepening concavity is present on the lateral surface (‘surangular fossa’). This depressed area terminates anteriorly to the level of the coronoid process. The coronoid process is prominent and bears a rugose texture. A small, anteroposteriorly oriented ridge is present directly posteromedially to the coronoid process and is likely part of the muscle attachment. The dorsal margin of the surangular forms a concave plateau posteriorly to the coronoid process. A promiment M.a.m.e process is present directly posteromedially to the plateau; this process points dorsomedially. The angular facet extends anteriorly up to the level of the coronoid process and covers the ventral half of the surangular posteriorly. This indicates the presence of an extensive angular, which is an ophthalmosaurid synapomorphy (int. al. Motani, 1999a; Druckenmiller et al., 2012; Fischer et al., 2012, 2014a; Zverkov & Prilepskaya, 2019). The teeth are peculiar in forming externally-visible plicidentine, texturing the acellular cementum ring and the root by very deep apicobasal grooves, as in Ichthyosaurus, Protoichthyosaurus, and Temnodontosaurus (McGowan, 1973; Godefroit, 1993a; Vincent et al., 2014; Brusatte et al., 2015; Lomax, Porro & Larkin, 2019). The root cross-section is oval as in all parvipelvians bar most platypterygiines (Fischer et al., 2012; Fischer, 2016) and the base of the enamel is easy to discern, which is usually the case in ophthalmosaurids (Fischer et al., 2016). Accordingly, we refer this specimen as Ophthalmosauridae indet. (Table 2).

MNHNL BM779 (Fig. 11) is a fragmentary partially articulated ichthyosaurian, containing parts of the skull, axial skeleton and scapular girdle, originating from the Rumelange locality. It belongs to Ophthalmosauridae, having a humerus with plate-like dorsal trochanter, a massive deltopectoral crest, a humeral facet for an anterior accessory element, a posterodistally deflected ulnar facet and a conspicuous acromial process on the scapula (Motani, 1999a; Fischer et al., 2012, 2013; Moon, 2017; Zverkov & Efimov, 2019) (Table 2). It resembles Arthropterygius in having an anteroposteriorly-short parietal symphysis, a deep ulnar facet on the humerus, a small humeral facet for an anterior accessory epipodial elements (Maxwell, 2010; Fernández & Maxwell, 2012; Zverkov & Prilepskaya, 2019). However, the presence of several autapomorphies on the cranial and appendicular elements indicates that this specimen constitutes a novel taxon, for which a dedicated manuscript is in preparation.

Figure 11 The new Humphresianum Zone ophthalmosaurid, early Bajocian, Luxembourg.

Selected anatomy of specimen MNHNL BM779. (A–C) right exoccipital in anterolateral (A), posteromedial (B) and posterior (C) views. (D and E) suraoccipital in posterior (D) and ventral (E) views. (F–H) left scapula in anterior (F), medial (G), and lateral (H) views. (I–K) right quadrate in lateral (I), anterior (J), and condylar (K) views. (L) right parietal in dorsal view. (M and N) right humerus in posterior (M) and dorsal (N) views. (O and P) right ulna in posterior (O) and dorsal (P) views.

PLESIOSAURIA De Blainville, 1835

CRYPTOCLIDIDAE Williston, 1925

Cryptoclididae indet.

MNHNL BM782 (Figs. 10A–10F) is a left plesiosaurian propodial from the Rumelange locality. The propodial is straight in the proximodistal direction and slightly curves dorsally. The shaft is fairly elongated. The dorsal tuberosity is wide and weakly demarcated from the capitulum; they form together an evenly rounded proximal surface. A median dorsal boss surrounded by muscle scars is present close to the distal end of the dorsal tuberosity. This boss is median (along the axis of the shaft) and thus differs from the anteriorly placed boss seen in the humeri of rhomaleosaurids and microcleidids (Owen, 1865; Smith & Benson, 2014). The propodial markedly expands distally, forming a preaxial flange and a longer postaxial flange. The preaxial flange forms a dorsoventrally-narrow triangular facet distally. The postaxial flange forms a semioval posterior facet that forms a marked angle (ca. 45°) with the ulnar/fibular. Both the radial/tibial and the anteroposteriorly- and dorsoventrally-shorter ulnar/fibular facets are slightly convex. The slightly longer radial facet forms a ca. 120° angle with the ulnar facet. However, a distal ridge is absent, unlike in Colymbosaurus (Benson & Bowdler, 2014; Arkhangelsky et al., 2019). The shape of the propodial with its fairly slender shaft and large preaxial and postaxial flanges indicate cryptocleidid affinities (Andrews, 1910a; Mehl, 1912; Brown, 1981; O’Keefe & Wahl, 2003; Knutsen, Druckenmiller & Hurum, 2012c; Roberts et al., 2017, 2020) (Table 2), notably resembling the humerus of humerus of Murænosaurus leedsi (Seeley, 1874b; Andrews, 1910a) and the femur of Tricleidus seeleyi (Owen, 1865).

Discussion

The Early–Middle Jurassic transition in marine reptiles, a state of the art

The very well-sampled (Benson et al., 2010) Lower Jurassic marine ecosystems of western Europe housed a vast menagerie of neoichthyosaurians, thalattosuchians, and plesiosaurians (Owen, 1860; Von Huene, 1922, 1931; McGowan, 1974, 1979; Benton & Taylor, 1984; Maisch & Matzke, 2000; O’Keefe, 2004; Großmann, 2007; Maisch, 2008, 2010; Benson, Evans & Druckenmiller, 2012; Martin et al., 2012; Bardet et al., 2014; Lomax & Massare, 2016; Johnson, Young & Brusatte, 2020). Evidence from body size, craniodental shape, and swimming capabilities suggests that these taxa occupied several niches within shallow marine ecosystems (Hauff, 1953; Massare, 1997, 1987, 1988; Böttcher, 1989; Godefroit, 1994a; McGowan, 1996; Buchholtz, 2001; Buchy, 2010; Fischer, Guiomar & Godefroit, 2011; Martin et al., 2012; Dick, Schweigert & Maxwell, 2016; Maxwell & Cortés, 2020). Such a diversity of forms made the Early Jurassic marine ecosystems of western Europe an iconic representation of Mesozoic marine life (Taylor, 1997).

Even though a drastic reduction of apparent diversity is expected following the lagerstätten effect of the early Toarcian localities (Benson et al., 2010), the Middle Jurassic assemblages, when well sampled (i.e., not before the Callovian), are markedly distinct from their Toarcian counterparts. For ichthyosaurians, the dominant and diversified early neoichthyosaurians (leptonectids, Suevoleviathan, Hauffiopteryx and Temnodontosaurus) are gone; the disappearance of non-thunnosaurian ichthyosaurians marks the end of large (>6 m) ichthyosaurian top predators (Massare, 1987), at least up until the Aptian, when some derived platypterygiines will presumably fill similar roles (Fischer et al., 2014b, 2016; Bardet, Fischer & Machalski, 2016; Fischer, 2016). Ichthyosaurians will never re-evolve carinated teeth like those seen in Temnodontosaurus platyodon and Temnodontosaurus trigonodon (Conybeare, 1822; Massare, 1987; Godefroit, 1993a), nor the sawfish-like morphology of Leptonectidae (Swinton, 1930; Von Huene, 1951; McGowan, 1986, 2003; Lomax, 2016). From the Callovian onwards, ichthyosaur assemblages overwhelmingly consist of ophthalmosaurids (but see Fischer et al. (2013)), up to the final extinction of ichthyosaurians (Fischer et al., 2016). While diverse ophthalmosaurid assemblages are known by the Kimmeridgian (Valenciennes, 1861; Druckenmiller et al., 2012; Arkhangelsky & Zverkov, 2014; Zverkov et al., 2015; Paparella et al., 2017; Moon & Kirton, 2018a; Delsett et al., 2019; Zverkov & Efimov, 2019; Barrientos-Lara & Alvarado-Ortega, 2020; Campos, Fernández & Herrera, 2020; Zverkov & Jacobs, 2020), only the closely related Ophthalmosaurus and Baptanodon as well as indeterminate but compatible forms are known in the Callovian–Oxfordian (Seeley, 1874c; Marsh, 1895; Gilmore, 1902, 1906; Knight, 1903; Andrews, 1910b; Arkhangelsky, 1999; Fernández & Iturralde-Vinent, 2000; Massare et al., 2006, 2014; Moon & Kirton, 2018b; Arkhangelsky et al., 2018; Otero et al., 2020).

This leaves a fairly long interval (late Toarcian–latest Callovian) for which the ichthyosaur fossil record is very scarce and geographically dispersed: Sander & Bucher (1993) reported a large cf. Stenopterygius from the late Toarcian of southern France; Maisch & Matzke (2000: p72) mention the presence of Temnodontosaurus in the late Toarcian of southern Germany; Vincent et al. (2013) reported Temnodontosaurus sp., and Stenopterygius-like forms in the late Toarcian of southern France and a coracoid bearing many similarities with MNHN DOU353 from the early Aalenian of southern France; finally, Brusatte et al. (2015) described Dearcmhara shawcrossi from the late Toarcian-Bajocian of Scotland. The only diagnosable ichthyosaur remains unambiguously known from Aalenian deposits is the holotype of Stenopterygius aalensis from Germany (Maxwell, Fernández & Schoch, 2012) (see also Arnaud, Monleau & Wenz (1976) and Maxwell, Fernández & Schoch (2012) for additional indeterminate specimens). Fragmentary ophthalmosaurid specimens have been reported in the Aalenian–Bajocian boundary of Argentina (Fernández, 2003) and in the early Bajocian of Argentina (Cabrera, 1939; Fernández, 1999; Gasparini et al., 2007; Fernández & Talevi, 2014) and Canada (Druckenmiller & Maxwell, 2014). A fragmentary rostrum from the early Bajocian of Argentina has been regarded as a representative of Stenopterygius (Cabrera, 1939) or as a Temnodontosaurus/Suevoleviathan-like form because of relatively long dental roots (Fernández & Talevi, 2014). However, this material lacks diagnostic features and is currently referred to as Ichthyosauria indet (Fernández & Talevi, 2014). It can be inferred from these occurrences that a replacement of early neoichthyosaurians by ophthalmosaurids took place across the Early–Middle Jurassic transition, but the generalized absence of Aalenian-Bathonian ichthyosaur fossils worldwide and Early Jurassic fossils in South America make it difficult to identify a precise turnover.

While less abundant than ichthyosaurians in early Toarcian Lagerstätten (Hauff, 1953), plesiosaurians already had also evolved a vast array of morphotypes by the Toarcian (O’Keefe, 2001b, 2001a; Benson, Evans & Druckenmiller, 2012; Smith & Araújo, 2017), presumably filling as many ecological niches: long-necked and small headed microclidids (Owen, 1865; Bardet, Godefroit & Sciau, 1999), gigantic, apex predatory rhomaelosaurids (Taylor, 1992; Cruickshank, 1994; Smith & Dyke, 2008; Smith & Vincent, 2010; Smith & Benson, 2014), and small-sized, moderately long-necked, and long-snout early pliosaurids (Benson et al., 2011; Ketchum & Benson, 2011a; Vincent, 2011; Fischer et al., 2017). Microcleidids supposedly go extinct after the Toarcian; their last definite record in the latest Toarcian of France (Sciau, Crochet & Mattei, 1990; Bardet, Godefroit & Sciau, 1999). However, an Aalenian specimen from France regarded as an indeterminate elasmosaurid by Vincent, Bardet & Morel (2007) possesses microcleidid features (eight-shaped cervical rib facet, lateral ridge on cervical centra) and might possibly be regarded as a member of that clade. Nevertheless, the long-necked morphotype is then colonised by cryptoclydids by the Callovian-Oxfordian (Andrews, 1909; Brown, 1981).

Only a couple of rhomaleosaurid lineages will survive up to the Middle/Late Jurassic, in high latitudes (Gasparini, 1997; Sato & Wu, 2008; Benson, Zverkov & Arkhangelsky, 2015). Thalassophonean pliosaurids concomitantly radiated and became the main marine apex predators from the Middle Jurassic to the early Late Cretaceous (Noe, 1999, 2001; Ketchum & Benson, 2011a; Benson et al., 2013; Benson & Druckenmiller, 2014; Fischer et al., 2017; Zverkov et al., 2018). A narrative where thalassophoneans filled (either competitively or passively) the niches previously occupied by Temnodontosaurus and derived rhomaleosaurids is sensible, but these animals have clearly distinct body plans and probably hunted differently (Massare, 1988; Buchholtz, 2001; O’Keefe, 2001a, 2002). In any case, there is evidence for the continuous presence of gigantic apex predatory plesiosaurians in Europe and elsewhere: Simolestes keileni from the upper Bajocian of eastern France (described by Godefroit (1994b), and currently under revision by S. Sachs), an indeterminate fragmentary jaw from the lower Bajocian of Switzerland (Sachs, Klug & Kear, 2019) and ‘pliosauroid’ tooth from the Aalenian-Bajocian of Australia (Long & Cruickshank, 1998; Kear, 2012). Whether these specimens are rhomaleosaurids or thalassophoneans obviously yields different outcomes as they might push the origin of apex predatory pliosaurids to the Bajocian, but these fossils nevertheless suggest that the niche of very large apex predators was not vacated for a long period, if at all.

Recent evidence revealed a series of basal metriorhynchoid crocodyliforms in the late Toarcian–Aalenian interval (Wilberg, 2015; Ösi et al., 2018; Aiglstorfer, Havlik & Herrera, 2020) suggesting an intense diversification across the Early–Middle Jurassic transition (Aiglstorfer, Havlik & Herrera, 2020), associated with a radiation into the pelagic realm (Cau & Fanti, 2011; Cau, 2014; Wilberg, Turner & Brochu, 2019). The effect of this radiation is evident by the late Middle and Late Jurassic, where abundant pelagic thalattosuchian taxa are known, occupying several ecological niches (Gasparini, Pol & Spalletti, 2006; Young & De Andrade, 2009; Young et al., 2012, 2014b; Herrera, Gasparini & Fernández, 2013; Foffa et al., 2017, 2018). Teleosauroids do not appear much affected by this transition however (Wilberg, Turner & Brochu, 2019), as many lineages and ecomorphotypes crossed this interval unscathed (Johnson, Young & Brusatte, 2020).

An abrupt turnover

In light of the short review above, it appears clear that fossiliferous successions in a geographically-restricted area are crucial to decipher both the tempo and the severity of the marine reptile turnover at the early–Middle Jurassic transition. The late Toarcian–Bajocian successions of Luxembourg and Belgium we analysed above yields novel biostratigraphic information that precise the turnover dynamics of ichthyosaurians and plesiosaurians during the Early–Middle Jurassic transition. In a global, broad-brush view of marine reptile macroevolution, the presence of rhomaleosaurids, microcleidids, and non-thunnosaurian ichthyosaurians in the late Toarcian of Luxembourg (Table 2) was expected, as these clades are already known to survive after the early Toarcian Lagerstätten (Bardet, Godefroit & Sciau, 1999; Vincent et al., 2013; Benson, Zverkov & Arkhangelsky, 2015). With the exception of possible remains from the Aalenian of France (Vincent, Bardet & Morel, 2007; see above), MNHNL DOU978 and the holotype of Microcleidus tournemirensis (Sciau, Crochet & Mattei, 1990; Bardet, Godefroit & Sciau, 1999) are the youngest-known microcleidid to date and indicate that the clade extended at least up to the very end of the Early Jurassic. However, recording microcleidids, large rhomaleosaurids, Temnodontosaurus-like forms, and more derived ichthyosaurians within the same basin that formed the early Toarcian Lagerstätten yields more important palaeobiogeographic implications. These fossils indicate that the main marine reptile clades of the Early Jurassic remained abundant and dominant in mid-latitude, epicontinental seas up to the end of the Toarcian at least. Although much less prolific than the underlying strata, the upper Toarcian fossil record suggest that no major turnover took place within the Toarcian as the structuration of the upper tier of marine ecosystems remained largely unchanged. The taxonomic diversity crash that follows the early Toarcian appears to be mainly a Lagerstätten effect, at least for marine tetrapods confirming previous suspicions (Benson et al., 2010; Maxwell & Vincent, 2015).

In stark contrast with the sense of continuity displayed by the upper Toarcian occurrences, the Bajocian fossil record of the Luxembourg area suddenly records the dominant clades of the Late Jurassic: cryptoclidids, ophthalmosaurids, thalattosuchians (Johnson et al., 2018), and probable pliosaurids (Godefroit, 1994b), with no evidence—so far—for the presence of more ancient clades. The specimen MNHNL BM782 is oldest-known cryptoclidid and MNHNL BM770 and MNHNL BM780–BM781 are two of the oldest ophthalmosaurids known, after a single partial forefin from the Aalenian– Bajocian boundary of Argentina (Fernández, 2003) and fragmentary basicranium from an equivalent of the Sauzei Zone in Canada (Druckenmiller & Maxwell, 2014), which directly underlies the Humphresianum Zone. The new occurrences we report here indicate that ophthalmosaurids rapidly dispersed, being almost simultaneously recorded in Canada, Argentina and Luxembourg by the early Bajocian.

The Belgo–Luxembourgian marine reptile record provides a clearer picture of the marine reptile turnover occurring at the Early–Middle Jurassic transition, indicating that this replacement is restricted to the sole Aalenian stage instead of possibly spanning the entire middle Toarcian–Bathonian interval. This reduces the uncertainty on the timing and duration of this turnover to 4 instead of 14 million years and packs a series of extinctions and diversifications within a short period of abrupt climate cooling and changes in oceanic currents (Korte et al., 2015). This Aalenian shift from a Toarcian Warm Mode to Aalenian–Bajocian Cool Mode (Korte et al., 2015) also appears associated with a marked faunal disruption in belemnites, where random (i.e., non-morphologically-selective) extinctions lead to a distinct drop in belemnite biodiversity at least in the northwestern Peri-Tethys Ocean (Dera, Toumoulin & de Baets, 2016; Neige, Weis & Fara, 2021). This major disruption in the evolutionary history of Jurassic belemnites ended at the Aalenian-Bajocian boundary and resulted in a radiation of the suborder Belemnopseina that partially replaced the previously dominant Belemnitina in the Western Tethys (Weis, Mariotti & Riegraf, 2012; Weis, Sadki & Mariotti, 2017), furthermore entailing a distinct Boreal vs. Tethyan belemnite provincialism (Doyle, 1987; Mariotti, Santantonio & Weis, 2007; Weis & Mariotti, 2007; Mariotti et al., 2012; Dzyuba et al., 2019). At the present state of knowledge, we can only speculate about whether faunal changes in belemnites and marine reptiles were independently impacted by the same factors, or even causally connected, since belemnites are an essential component of the food spectrum in some marine reptiles (Massare, 1987; Böttcher, 1989; Dick, Schweigert & Maxwell, 2016), their faunal change could have triggered a disruption of the trophic chain up to the giant top predators.

Conclusions

A generalised turnover affected marine reptile communities across the Early–Middle Jurassic transition. However, the extremely poor fossil record of the middle Toarcian-Bathonian interval leaves a ca. 14 million years window of the uncertainty for this important event. Our thorough analysis of the ichthyosaurian and plesiosaurian record of the late Toarcian to Bajocian successions in a confined palaeogeographic setting (the Belgo–Luxembourgian sub-basin) indicates that:the structuration of the upper tier of marine ecosystems of Western Europe remained unchanged up to the very end of the Early Jurassic, with the presence of large rhomaleosaurid plesiosaurians, microcroclidid plesiosaurians, as well as Temnodontosaurus-like and baracromian ichthyosaurians in the late Toarcian.

the dominant clades of the Late Jurassic, cryptoclidid plesiosaurians and ophthalmosaurid ichthyosaurians, arose and dispersed earlier than expected, being now recorded in the early Bajocian of Luxembourg.

the marine reptile turnover of the Early–Middle Jurassic transition is more abrupt than previously supposed, being restricted to the sole Aalenian stage, that is, 4 million years in the Belgo–Luxembourgian sub-basin.

Supplemental Information

Supplemental Information 1 R script for plotting PBDB collections and centra proportions.

Click here for additional data file.

Supplemental Information 2 PDBD collections data.

Marine reptile collections through the Jurassic.

Click here for additional data file.

Supplemental Information 3 Jurassic temporal data.

Binned at the stage level.

Click here for additional data file.

Supplemental Information 4 Centra proportions.

Click here for additional data file.

We thank the curatorial staff of the IRSNB for their help. We also thank the main contributors of the Paleobiology Database data we used: Prof. R. B. J. Benson, Dr. M. T. Carrano, Dr. J. P. Tennant, Dr. H. P. Street, Prof. R. J. Butler, Prof. M. Clapham and Prof. M. Uhen. We warmly thank We warmly thank Dr. M. Trotta for her help in creating the geographic map, reviewers Prof. M. Fernández, Dr. A. Roberts, and Dr. D. Foffa, and editor Dr. M. T. Young for their insightful comments and constructive suggestions.

Institutional abbreviations

IRSNB Royal Belgium Institute of Natural Sciences, Brussels, Belgium

MNHNL Muséum national d’histoire naturelle du Luxembourg, Luxembourg-ville, Luxembourg

ULg-PA Collections de paléontologie animale de l’Université de Liège, Liège, Belgium

Additional Information and Declarations

Competing Interests

Author Contributions

Data Availability

The authors declare that they have no competing interests.

Valentin Fischer conceived and designed the experiments, performed the experiments, analyzed the data, prepared figures and/or tables, authored or reviewed drafts of the paper, and approved the final draft.

Robert Weis conceived and designed the experiments, prepared figures and/or tables, authored or reviewed drafts of the paper, provided essential biostratigraphic data, and approved the final draft.

Ben Thuy conceived and designed the experiments, authored or reviewed drafts of the paper, provided essential biostratigraphic data, and approved the final draft.

The following information was supplied regarding data availability:

R script, raw measurements, the Paleobiology database data, and temporal data are available as Supplemental Files.

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
