# Peer review of "Refining the marine reptile turnover at the Early–Middle Jurassic transition"

_PeerJ, doi:10.7717/peerj.10647_

## Round 0.1 · original submission · Minor Revisions

Dear authors,

I have accepted the reviewer recommendations of ‘minor revisions’.

Please note all three reviewers have made different comments. Please consider the comments of reviewer two: vertebral analysis and PDB limitations.

I look forward to receiving your revised manuscript.

·

Basic reporting

As a not native speaker, I am not qualified to evaluate the English (grammar, spelling, etc). However, I would like to point out that it was very clear to read it all throughout. The structure of the article is correct and the figures and tables relevant, as well as literature cited (I only made some minor few suggestions, see below). The Introduction section provides a solid background and introduced properly to the main issue of the manuscript: the putative early Middle Jurassic marine herpetofaunas turn-over“. Despite being a major biotic event (especially for the understanding of the marine Mesozoic diversity) is poorly understood and also, somehow, overlooked. First pointed out by in the pioneer contributions of Massare (1987) and Bardet (1994), Aalenian-Bajocian interval was crucial for the re-organization of marine tetrapod communities however few (or almost none recent contributions) contributions deal with this topic. Within this framework this contribution is welcome. Marine reptiles from this interval are also scarce worldwide. Based on the detailed description of fossils from successive formations of a single sub-basin from Belgium-Luxembourg, the authors restrict the previously hypothesized turnover to the Aalenian. They also provide hypotheses on environmental changes that could have driven it. In this sense, this article will be of interest to paleontologists working on marine reptiles but, in a broader context, it will be useful to understand the evolution of marine ecosystems during the Mesozoic. Also interesting is that this article provides an interesting starting point for the further test outside de European basins to test if the early Middle Jurassic event was locally restricted or worldwide

Experimental design

no comment

Validity of the findings

no comment

Additional comments

Comments and suggestions (also annotated on the pdf): on the Description section, I suggest indicating the specific part of the figure where the specimen is shown (e.g. Figure 6 A, B). This could help the reader particularly when two or more specimens are referred to in the same figure (e.g. figures 5-7, 10-11)
Line 717: I suggest including, as part of kimmeridgian ophthalmosaurids from North America, the recently described Acuetzpalin from Mexico (which it is not closely related to Baptanodon or Ophthalmosaurus). Barrientos Lara and Alvarado-Ortega (2020). Journal of South American Earth Sciences 98, 102456

Line 720: for the age of Baptanodon materials, see Massare et al. 2013. Geological Magazzine:1-16. I think that most of the materials referred to Baptanodon have been recovered from the Oxfordian of the Sundance Formation.

Line 738: Maybe it would be more useful (instead of Fernández, 1999) including the citation of Fernández & Taleli (2013) as it is mentioned that during the early Bajocian in the paleogulf represented by the Neuquen Basin in Patagonia there were at least three different forms: Molllesaurus, Chacaicosaurs + "Temnodontosaurus-Suevoleviathan like form described originally as Stenopterygius grandis Cabrera, 1939 [CABRERA, A. 1939. Sobre un Nuevo ictiosaurio del Neuquén. Notas del Museo de La Plata 21, 485–91.].

·

Basic reporting

The introduction/background section is through and literature is well referenced. I have noted some clarifications regarding the research question in the annotated pdf. Although present in the introduction the research question could be presented more clearly. The results tie in well with gap in knowledge highlighted.

The English used is at a high standard, I have limited corrections here (see pdf).

The structure, as far as I can tell conforms to the PeerJ standard as per the author guidelines. The figures are relevant, are of high quality and clearly annotated. Raw data is supplied.

Experimental design

The research is within the scope of the journal and believe it fits well with other articles published in PeerJ. The research question is present in the introduction, but could be presented more clearly. Use your clearly stated bulletin points in the conclusion to present your question(s) clearly in the introduction. They are present, but not as clearly defined as they could be. This research does fit a knowledge gap.

For description sections I have a few comments noted in the annotated PDF. There are a few characters that are used here, which are no longer regarded as diagnostic or have questionable validity. The laterally compressed tail vertebrae in thunnosaurian ichthyosaurs is one (see pdf) and the distal ridge on the propodial of Colymbosaurus is another.

For Colybosaurus I suggest that this feature is only valid for C. megadeirus as suggested in Roberts et al., 2017 (https://doi.org/10.1080/02724634.2017.1278381). However this distal ridge can form during taphonomic compression, varies in ontogeny and should be used with care. You can use it for Colymbosaurus, but also add in the caveat that it may form during taphonomic compression and that its validity has been questioned.

Otherwise, the methods sections requires additional details to be added. Include the number of samples used for your analyses in this section. How were the vertebrae measured? Did you include all pbdb data available for your stated groups? Which taxonomic level did you use? Did you check for mistakes or bias? Also where is the taxon information in your dataset (ie. Which occurrence belongs to which group?), this is not included in the provided dataset. Previously I have noticed that some of the timebins in pbdb have not be accurate and followed an older version of the geological timescale for older entries. I didn’t locate any mistakes when I checked this in your dataset, but might be worth double checking.

Validity of the findings

Underlying data is provided, although additional information is required as stated above. Conclusions are clearly stated, but require better linking to the introduction.

My speculations here are: I am not convinced that there is sufficent information to justify the vertebral analysis. This requires additional support and evidence from the literature. Ichthyosaur vertebrae vary to some degree in their ratios depending on where they are from in the vertebral column. Plesiosaurs even more so. If you more specific in comparing different regions of the skeleton ie: middle cervical centra for plesiosaurs of proximal caudals for ichthyosaurs it would make more sense to compare them. Right now, with this limited dataset, I am not sure if it strengthens your argument, or is even required.

Additional comments

I greatly enjoyed reviewing the manuscript titled “Refining the marine reptile turnover at the Early-Middle Jurassic transition, by Fisher et al. I suggest minor revisions prior to publication as additional information needs to be included in the introduction and methods sections, in addition to a few changes elsewhere in the manuscript. Otherwise an interesting and valuable article, highlighting a time-interval poorly known, but important for marine reptile evolution. I have submitted additional questions and comments in an annotated PDF.

·

Basic reporting

I find this manuscript well-written, clear, unambiguous. The relevant literature is appropriately referenced, and the introduction, description and discussion are richly complimented with context.
I suggested some improvement for the figures and figure-referencing (see below and attached PDF).

Experimental design

All descriptions, comparisons are rigorous, valid, and clearly fills a well-defined knowledge gap. Methods and results are reported in details.

Validity of the findings

All results are robust. I particularly appreciated the attitude that the authors took when describing several specimens. I think that the authors did an exceptional work in their identifications and in supporting them, while at the same time they did not shy away from highlighting similarities with specific taxa.

Additional comments

Dear Authors and Editor,
This manuscript describes a series of fragmentary specimens (plesiosaurians and ichthyosaurians) from the Early-Middle Jurassic of Luxemburg and their implication for marine reptile evolution. This interval is poorly understood and is sandwiched between the well-known and iconic Lagerstatten of the Early and late Middle Jurassic. Interestingly the marine tetrapod fauna changed considerably during this interval. As thus any specimen found in between is bound to fill in a ~14 million year gap in marine reptile evolution. And this is exactly what the main achievement of this manuscript is.
I do appreciate the clear exposition, strong background and referencing of all relevant literature – except perhaps that some additional mention to the thalattosuchians found in the area would be welcome (but see the annotated PDF for further details).
The description and comparisons are nothing short but excellent. Arguably the strongest point of the manuscript: they are precise, complete and the identifications of all specimens are well supported by clear references to diagnostic features. In this regards I do appreciate that the authors adopted a cautious approach and referred the specimens to higher taxonomy ranks, but also did not refrain to point out similarities with genera and species.
The results from these sections are put in context in the discussion that I also found clear and particularly enjoyable. Once again thalattosuchians are hardly mentioned – and even though teleosauroids went through less dramatic changes, this is roughly the time when metriorhynchoids diversify and become fully pelagic. Of course the authors do not have evidence to add to this process, but when studying faunal turnover of this interval, this is an evolutionary event that should at very least be noted. Again, please refer to the annotated PDF for additional details.
While my opinion of the manuscript is largely positive there are a few issues that I would like to point out.
1. First and foremost the description and figures.
I find the figures and plate excellent. But there are serious problem in how they are organised and cited in the text.
Many different specimens are described in the manuscript and only some of them are figured. I understand that not all specimens can figured and that selection should be made. However, while I am ok for undiagnostic specimens to be left out of the plates, I thin think that specimens which have important diagnostic features should be all figured.
The figures and panels should appear in the order they are mentioned - this is not currently the case. For instance specimens in figure 8 are described before elements depicted in figure 7. This principle should also apply to sub-figure (e.g. elements depicted in figure 5A should be described before those in Figure 5B). Please make sure that all orders are respected.
Another element of confusion is that some specimen numbers are left out the figure caption. This made it very difficult to understand which description referred to which figure.
I the description section the only in-text reference of figures is a line after the systematics table, but as previously mentioned, not all described specimens are actually figured, which is very confusing in my opinion. While reading the specimen description I often looked for the figures but the lack of precise in-text figure references ( = reference to specific parts of the figures i.e. “Figure 3A-B”), and lack of clarity of what specimen is actually in the figures made it all very difficult.
I would like to suggest a couple of solution for these issues:
- Many many more in-text references should be added. Please add in text figure reference (E.g. “Figure 5A”) when describing each element/specimen.
- Also important, make also sure that the specimens are cited in the right order. For instance the specimens in Figure 5 are confusingly cited out of order. The angular in Fig5G-H is described after the quadrate Fig. 5I-K, which is described before the vertebrae (Fig AA-D). Please re-organise the figures/text so that the same specimen is depicted together, and they are described in order.
- All figured specimens numbers should appear in figure captions (e.g. MNHNL DOU978 is not mention in the caption of Fig7).
- I’d suggest to add in “boxes” representing the same taxon (or group) in figures that include multiple taxa (e.g. Fig. 7 include Rhomaleosauridae and sp. Microclidus; but also see Fig. 9, 10)
- As mentioned, I do not expect all specimens to be illustrated. However it should be made clearer which one are and which are not. Adding a column to table 2 could help with this.
Please increase the font size in all figures: the smallest text is very very small (I’d say at least double it).
2. Quantitative study.
In support of their descriptions the authors add a quantitative analysis of the proportion of ichthyosaurian centra. The point of this section should be better explained in the method section.
I do appreciate the effort that the authors put to quantitatively prove their point, however, while the results confirm a reduction in disparity through the selected interval, I am not sure it really adds much to the manuscript. I think that the same point comes across nicely through the comparison of the ichthyosaur specimens. It is always nice seeing quantitative support to qualitative data, but in this case the centra-proportion analysis could be cut out of the manuscript without compromising or undermining the results. I leave this decision to the authors.
A series of minor comments and clarifications are included in the attached PDF.
Upon minor modifications, some clarification and addressing of the above mentioned issues (please also refer to PDF for more details), I consider this manuscript ready for publication.
I thank the Editor and Authors for the opportunity to review this manuscript and I look forward to seeing it brought to the next stages of publication.
If appropriate and allowed by PeerJ regulations, I will be happy to be privately contacted by the Authors for additional clarifications on my comments or general questions regarding this review.
Many thanks,
Davide Foffa
RCE 1851 Postdoctoral Fellow - National Museums Scotland
d.foffa@nms.ac.uk OR davidefoffa@gmail.com)

---

## Round 0.2 · accepted · Accept

Dear authors,

I have accepted the reviewer's decision of 'accept'.

You will shortly be contacted by the production staff, who will take you through the proof-stage.

Congratulations, and I hope you will consider using PeerJ as your publication venue in future.

·

Basic reporting

The edited sections of the paper have a high standard of English, and I have found not grammatical or spelling errors.
The other reviewers have suggested additional literature which has been included, which strengthen the context.
The new layout for the figures is improved. I have one question about Figure 3C, where part of the image looks cut off. This should be checked before final submission.

I am satisfied with the changes to the introduction to tie up with the conclusion.

Experimental design

The research is within the scope of the journal and believe it fits well with other articles published in PeerJ. The changes to the introduction are sufficient and this paper clearly fills a knowledge gap.
My previous comments on the methods are taken into account and I see no further issue.

Validity of the findings

Underlying data is provided and sufficient explanation is given to allow replication. The conclusions are well stated and link to the introduction.
I think the removal of the vertebral analysis was well founded as it was not required to strengthen your argument.

Additional comments

All my comments and suggestions have been taken into consideration or included.
I have 3 suggestions/comments that I would recommend before publication:
Line 69 - Change "tempo" to "duration"
Line 221-222 - Be aware that propodials in Colymbosaurus megadeirus can be massive. Significantly larger than any other cryptoclidid. So this may not be best comparison.
Figure 3C - Make sure this image has not been cut.

As these are only suggestions and not hard changes, I am happy to accept this paper for publication.